# Efficient variational simulation of non-trivial quantum states

**Wen Wei Ho[1⋆] and Timothy H. Hsieh[2,3]**

**1** Department of Physics, Harvard University, Cambridge, Massachusetts 02138, USA
**2** Kavli Institute for Theoretical Physics, University of California,
Santa Barbara, California 93106, USA
**3** Perimeter Institute for Theoretical Physics, Waterloo, Ontario N2L 2Y5, Canada

⋆ wenweiho@fas.harvard.edu

## Abstract

We provide an efficient and general route for preparing non-trivial quantum states that are not adiabatically connected to unentangled product states. Our approach is a hybrid quantum-classical variational protocol that incorporates a feedback loop between a quantum simulator and a classical computer, and is experimentally realizable on near-term quantum devices of synthetic quantum systems. We find explicit protocols which prepare with perfect fidelities (i) the Greenberger-Horne-Zeilinger (GHZ) state, (ii) a quantum critical state, and (iii) a topologically ordered state, with $L$ variational parameters and physical runtimes $T$ that scale linearly with the system size $L$. We furthermore conjecture and support numerically that our protocol can prepare, with perfect fidelity and similar operational costs, the ground state of every point in the one dimensional transverse field Ising model phase diagram. Besides being practically useful, our results also illustrate the utility of such variational Ansätze as good descriptions of non-trivial states of matter.

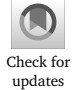

# 1   Introduction

Recent experimental advances in designing and controlling well-isolated synthetic quantum systems of many-particles, such as trapped ions [1, 2], cold atoms [3, 4], superconducting qubits [5, 6], etc., have allowed for the study of a plethora of interesting physical phenomena. These include topological order [7–9], phase transitions [10,11], thermalization [12,13], and time crystals [14, 15]. Equally exciting is the potential to use these platforms for performing quantum simulations and computation [4, 6, 16], or for speed-ups in quantum metrology precision measurements [17–20]. For such studies and the implementation of quantum information protocols, the preparation of complex quantum many-body states, i.e. those with non-trivial patterns of entanglement that are not adiabatically connected to short-ranged entangled states, is vital. For instance, topological states have long-range, non-local patterns of entanglement, and the Greenberger-Horne-Zeilinger (GHZ) state is an essential resource in quantum many-body metrology measurement proposals and has an entanglement pattern that is many-body in nature [17–20]. Furthermore, measurement-based quantum computing requires highly entangled initial states [21–23]. Therefore, it is important to have generic, explicit, resource-efficient schemes for preparing non-trivial quantum states.

In this paper, we demonstrate the efficient preparation of certain non-trivial states of interest, using a variational, hybrid quantum-classical simulation, which utilizes the resources of a quantum simulator and a classical computer in a feedback loop. In short, given Hamiltonians or gates (quantum resources) realizable in a quantum simulator, a quantum state $|\psi(\boldsymbol{\gamma}, \boldsymbol{\beta})\rangle$ is produced, with $(\boldsymbol{\gamma}, \boldsymbol{\beta}) \equiv (\gamma_1, \cdots \gamma_p, \beta_1, \cdots, \beta_p)$ parameterizing a finite set of $2p$ variational angles (or times) that the Hamiltonians or gates are run for. A cost function, usually taken to be the energy of some target Hamiltonian, is then evaluated within the resulting state and optimized for in a classical computer, which yields a new set of $2p$ angles to be implemented to be fed back into the quantum simulator. The entire process is iterated and the simulation terminates when the cost function has been desirably optimized (see Fig. 1); in this way, a good approximation to the ground state of the target Hamiltonian according to the cost function is then produced.

Such variational quantum approaches have been developed and utilized in a number of contexts, such as in quantum chemistry [24, 25], and also in classical optimization problems (for example, as the 'Quantum Approximate Optimization Algorithm' [26, 27]), with recent experiments demonstrating its success in platforms like photonic quantum processors [24], and programmable, analog quantum simulators of trapped ions [28]. There are a number of

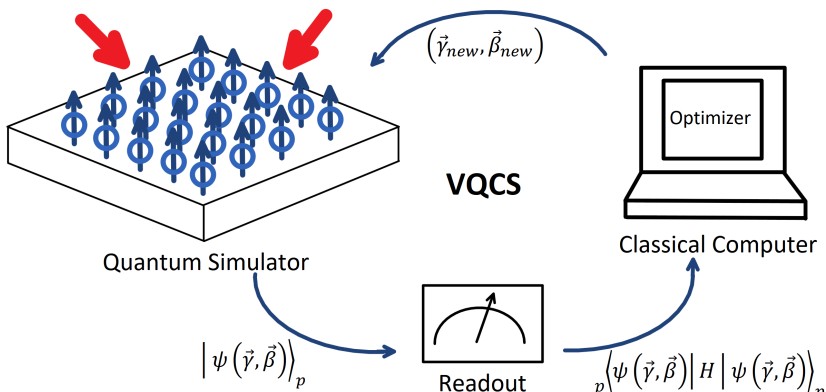

Figure 1: Schematic depiction of the hybrid variational quantum-classical simulation (VQCS) used to target non-trivial quantum states. A quantum simulator of e.g. trapped ions is used to unitarily prepare a target state $|\psi(\boldsymbol{\gamma}, \boldsymbol{\beta})\rangle_p$ via the protocol (5) using a set of variational angles $(\boldsymbol{\gamma}, \boldsymbol{\beta})$ of some fixed number of iterations $p$. A cost function, for e.g. the global energy of some target Hamiltonian $H_T$, $_p\langle\psi(\boldsymbol{\gamma}, \boldsymbol{\beta})|H_T|\psi(\boldsymbol{\gamma}, \boldsymbol{\beta})\rangle_p$ is then measured (achievable due to single-site resolution in measurements of quantum simulators). The result is then fed into a classical computer, which finds the next set of angles that optimizes the cost function. The simulation terminates when the cost function is desirably optimized; the resulting quantum state is then the near to the target state.

properties which make the variational quantum simulation (described in brief above, and in more detail later), appealing [29, 30]: it can be run on any quantum device, such as a digital quantum simulator, i.e. a gate-based universal quantum computer, or a (possibly tuneable) analog quantum simulator, in which the interactions between qubits are dictated by those in a given physical platform [30]. Furthermore, the very nature of the protocol makes it well suited for implementation in current quantum simulators of synthetic quantum systems, leveraging upon the tunability and single-site resolution of measurements in these platforms. In fact, the feedback loop of the protocol allows one to mitigate systematic errors that might be present in the experimental setups.

Employing this variational quantum-classical simulation and with local, uniform Hamiltonians, we show here that such protocols can be used to target the GHZ state, the critical state of the 1d transverse field Ising model (TFIM), and the ground state of the 2d toric code, all with *perfect fidelities*, using $2p = L$ variational parameters, and with minimum runtimes $T$ that scale linearly with the system size, $T \sim L$, where $L$ is the linear dimension of the systems. We furthermore conjecture, and support with numerical data, that the entire ground state phase diagram of the TFIM can interestingly be produced with perfect fidelities and similar operational costs. Lastly, as an additional study, we consider preparing the ground states of antiferromagnetic (AFM) Heisenberg chains, and find that the protocol is able to achieve them efficiently and with very good fidelities.

This concretely demonstrates the ability of variational quantum-classical protocols to efficiently and unitarily prepare a variety of quantum states with non-trivial patterns of entanglement, and also illustrates the utility of such ansätze as good descriptions of non-trivial states of matter. Note that for the Ising model and toric code, explicit circuits for the ground states are known, for example in terms of a unitary circuit for the 1d XY-model, exploiting its free fermion nature and fundamentally based on the Fourier transform [31], and in terms of a tensor network representation for the toric code [32]. However, such circuits involve nonuni-

form applications of multiple types of gates, and are not very practically realizable, especially in analog near-term quantum simulators [30]. This is in contrast to our method, which only requires time evolution between simple, uniform local Hamiltonians, and thus provides physically realizable roadmaps for quantum state preparation.

## 2 Non-trivial quantum states

Let us start by expounding upon the non-trivial nature of certain quantum states that we are interested in. Consider a target state $|\psi_t\rangle$ and an unentangled product state $|\psi_u\rangle$, both defined on a system with linear dimension $L$. $|\psi_t\rangle$ is said to be non-trivial if there does not exist a local unitary circuit $U$ of finite depth (i.e. scaling as $O(L^0)$) that connects the two: $|\psi_t\rangle = U|\psi_u\rangle$ [33]. Instead, the depth of a local unitary circuit connecting the two must be at least $O(L^\alpha)$ with $\alpha > 0$. Intuitively, nontrivial states have entanglement patterns fundamentally different from product states. While this is a statement made at the level of the wavefunction, from the perspective of local Hamiltonians and gaps, such states are separated from product states by a gap-closing phase transition in the thermodynamic limit, and thus preparing them with, for example, the quantum adiabatic algorithm [34, 35] is hard.

We now review why the GHZ, critical, and topologically ordered states are nontrivial. Consider first the GHZ state,

$$|\text{GHZ}\rangle \equiv \frac{1}{\sqrt{2}}(\otimes|Z = 1\rangle + \otimes|Z = -1\rangle), \tag{1}$$

where $X, Z$ are Pauli operators. Suppose there exists a local, finite-depth unitary $U$ that takes the completely polarized product state $|+\rangle \equiv \otimes|X = 1\rangle$ to $|\text{GHZ}\rangle$. Due to locality, there exists a Lieb-Robinson bound which limits the spread of information and entanglement under this evolution, implying that $U$ can only generate a finite correlation length $\xi$ for the final state. Measuring a long-range spin-spin correlator gives

$$\langle\text{GHZ}|Z_i Z_j|\text{GHZ}\rangle = 1, \tag{2}$$

while on the other hand the same quantity can be expressed as

$$\langle+|U^\dagger Z_i U U^\dagger Z_j U|+\rangle, \tag{3}$$

which in the limit $|i - j| \gg \xi$ decomposes as

$$\langle+|U^\dagger Z_i U|+\rangle\langle+|U^\dagger Z_j U|+\rangle = \langle\text{GHZ}|Z_i|\text{GHZ}\rangle\langle\text{GHZ}|Z_j|\text{GHZ}\rangle = 0, \tag{4}$$

a contradiction. Similar arguments apply to critical states which have power-law correlations, and topologically ordered states which have long-range correlations in loop operators and non-zero topological entanglement entropy [36–39].

## 3 Variational Quantum-Classical Simulation (VQCS)

We define below the variational quantum-classical simulation (VQCS), which involves utilizing the resources of both a quantum simulator and a classical computer in a feedback loop for the purpose of preparing a non-trivial quantum state of interest. It is motivated by the Quantum Approximate Optimization Algorithm (QAOA) [26, 27] and variational quantum eigensolver (VQE) algorithms [25, 28].

The protocol is envisioned to be run on a quantum simulator that can realize certain interactions between qubits and single qubit rotations, which we denote schematically by $H_1, H_2$ (these will be specified explicitly in the examples considered). On an analog quantum simulator, the interactions are the ones realizable in a given physical platform; while on a digital quantum simulator, in principle any interactions can be simulated given access to a universal gate set $G$. The aim is to produce a good approximation to the ground state of a target many-body Hamiltonian $H_T$, which we will assume to be a linear combination of $H_1$ and $H_2$. The VQCS begins with the ground state of $H_1$, which we denote $|\psi_1\rangle$, and evolves the state with the following sequence alternating between $H_2$ and $H_1$:

$$|\psi(\boldsymbol{\gamma}, \boldsymbol{\beta})\rangle_p = e^{-i\beta_p H_1} e^{-i\gamma_p H_2} \cdots e^{-i\beta_1 H_1} e^{-i\gamma_1 H_2} |\psi_1\rangle. \tag{5}$$

For a fixed integer $p$, there are $2p$ variational angles (or times) $(\boldsymbol{\gamma}, \boldsymbol{\beta}) \equiv (\gamma_1, \cdots \gamma_p, \beta_1, \cdots, \beta_p)$. Note that for a digital simulator, the unitaries $e^{-i\beta_i H_1}$, $e^{-i\gamma_i H_2}$ would have to be decomposed using the gates in $G$. We cannot address this decomposition in full generality, but will do so for the transverse field Ising example presented shortly. We call such a protocol VQCS$_p$.

A cost function $F_p(\boldsymbol{\gamma}, \boldsymbol{\beta})$, such as the energy expectation value of the target Hamiltonian

$$F_p(\boldsymbol{\gamma}, \boldsymbol{\beta}) = {}_p\langle \psi(\boldsymbol{\gamma}, \boldsymbol{\beta})| H_T |\psi(\boldsymbol{\gamma}, \boldsymbol{\beta})\rangle_p, \tag{6}$$

is then evaluated, which possibly involves a rotation into the appropriate basis in order to measure individual expectation values. A classical computer then performs an optimization to produce a new set of angles $(\boldsymbol{\gamma}, \boldsymbol{\beta})$, which are then fed back into the quantum simulator and the process repeated till the cost function is desirably minimized. The state corresponding to these optimal angles is therefore the optimal state that can be prepared by the protocol given this cost function. As the VQCS is envisioned to be run on near-term quantum simulators which are inherently noisy (so called 'Noisy, Intermediate-Scale Quantum' (NISQ) technology [30]), the physical runtimes $t = \sum_i^{p=L/2} (\gamma_i + \beta_i)$ of the VQCS constitute an important measure of the feasibility of the protocol – in general, shorter runtimes lead to less noise encountered and a better implementation.

It is clear that the optimal solution from VQCS$_{p+1}$ is always better than VQCS$_p$'s. Moreover, for large $p$ the VQCS can approximate a quantum adiabatic algorithm (QAA) of the form $H(t) = f(t)H_1 + (1-f(t))H_2$ for any smooth function $f(t)$ via Trotterization. Thus, ground states of target Hamiltonians of the form $H_T = H_1 + gH_2$ for some parameter $g$ can always be achieved in VQCS as $p \to \infty$, since QAA can produce arbitrary accuracy the target ground state of $H_T$ for any finite-size system if the speed of traversal is vanishingly small. However, for all practical purposes, the correspondence between the VQCS and QAA at small $p$ is not so clear, and thus in what follows we explore how well the VQCS can target certain hard-to-prepare quantum many-body states.

# 4 Using VQCS to prepare nontrivial quantum states

We demonstrate here that the efficacy and efficiency of VQCS in preparing the following states: (i) the GHZ state, (ii) a quantum critical ground state, (iii) a topologically ordered state, and (iv) the ground state of an antiferromagnetic (AFM) Heisenberg chain.

## 4.1 GHZ state

The GHZ state is defined in Eq. (1) and can be taken to be the ground state of the following target Hamiltonian,

$$H_T = -\sum_{i=1}^{L} Z_i Z_{i+1}, \tag{7}$$

in the symmetry sector $S = \prod_i^L X_i = 1$. For simplicitly we assume periodic boundary conditions.

We choose in this case $H_1 = -\sum_i X_i$ with the product ground state $|\psi_1\rangle = |+\rangle = \bigotimes_i^L |+\rangle_i$, where $X_i |+\rangle_i = |+\rangle_i$. $H_1$ has a straightforward implementation in both the analog and digital settings. We choose $H_2 = H_T$; on a digital quantum simulator, this can be achieved using elementary two-site gates $Z_i Z_{i+1}$:

$$e^{-i\gamma H_2} = \prod_{i=1}^{L/2} e^{-i\gamma Z_{2i} Z_{2i+1}} \prod_{i=1}^{L/2} e^{-i\gamma Z_{2i-1} Z_{2i}}, \tag{8}$$

such that each unitary $e^{-i\gamma H_2}$ in the VQCS protocol can be considered as a depth-2 quantum circuit, so that overall the VQCS$_p$ can be realized as a quantum circuit with depth at most $3p$, see Fig. 2 and [40] for a related discussion. On an analog simulator, $H_2$ can be approximately realized, for example as the Ising interactions $\sum_i \frac{1}{|i-j|^\alpha} Z_i Z_j$ that occur naturally in trapped ion ($\alpha \in (0,3]$) or neutral Rydberg atom ($\alpha = 6$) quantum simulators, for large $\alpha$.

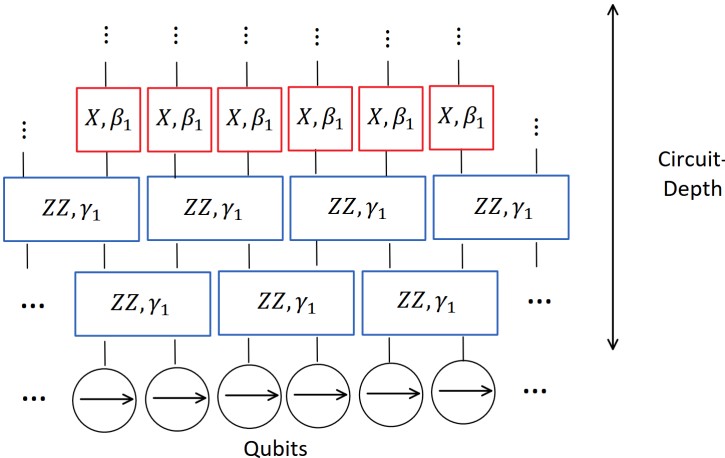

Figure 2: Quantum circuit realization of the VQCS protocol on a digital quantum simulator. Shown is the first layer of the protocol (5) using angles $(\gamma_1, \beta_1)$ corresponding to time evolution by two-site and one-site gates $Z_i Z_{i+1}$ and $X_i$ respectively. Subsequent layers utilize different angles $(\gamma_n, \beta_n)$. For a total of $p$ layers corresponding to VQCS$_p$, the quantum circuit has a depth of at most $3p$.

If we start with the polarized state $|+\rangle$ which has $S = +1$, then since the VQCS protocol (5) respects this symmetry, optimization of (6) as $p \to \infty$ will yield the GHZ state, with $\lim_{p\to\infty} F_p(\gamma, \beta)/L \to -1$. We implement the VQCS, finding numerically the optimal angles $(\gamma_*, \beta_*)$ that minimize (6) via a search by gradient descent of the parameter space $\gamma_i, \beta_i \in [0, \pi/2)$ for all $i$, for system sizes $L \leq L_1 = 18$, and for $p \leq L_1/2$. We restrict each angle to be any contiguous interval of length $\pi/2$ because $e^{-i\frac{\pi}{2} H_2} \propto 1$ and $e^{-i\frac{\pi}{2} H_1} \propto S$; furthermore, in order to give the angles $(\gamma, \beta)$ an interpretation of 'time', we choose $\gamma_i, \beta_i \in [0, \pi/2)$.

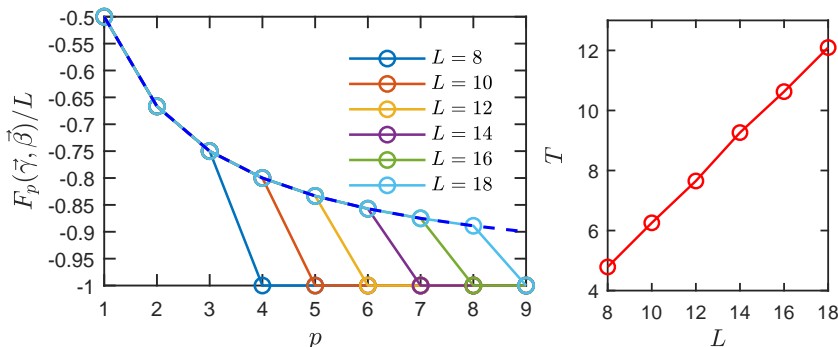

Figure 3: Preparation of GHZ state. (Left) Optimal cost function (6). One sees that $F_p(\gamma,\beta)/L = -1$ for $p \geq L/2$; in other words, the GHZ state is created with perfect fidelity using $\text{VQCS}_{p\geq L/2}$. We have also plotted a conjectured analytic expression $-p/(p+1)$ (from [26]) for the optimal cost function as the dashed blue line. (Right) Total minimum time $T = \min_{(\gamma,\beta)}\left[\sum_i^{p=L/2}(\gamma_i + \beta_i)\right]$ required for the VQCS to produce the GHZ state with perfect fidelity using $\text{VQCS}_{p=L/2}$. The minimization is performed over all the numerical solutions found. One sees a linear trend $T \sim L$.

We note that, for fixed $L$, assuming a fine mesh of each interval $[0,\pi/2)$ into $\mathcal{M}$ points, a brute force search of this parameter space takes an exponentially long time $t \sim O(\mathcal{M}^{2p})$ in $p$. Consequently, we have ensured that the total number of runs performed is large enough to ensure convergence of the search algorithm to the global minimum.

Fig. 3 shows the results. We see that interestingly, the GHZ can be prepared with *perfect* fidelity, to machine precision, using the protocol $\text{VQCS}_{p^*}$, with $p^* = L/2$. We note that there are multiple optimal solutions for $(\gamma,\beta)$ that give this perfect fidelity (furthermore, the vector of angles is symmetric under the reflection $\gamma_i \longleftrightarrow \beta_{L-i+1}$; this is due to the Kramers-Wannier duality of the Ising model which relates the paramagnet (product state) and the ferromagnet (GHZ)). Since each angle $\gamma_i, \beta_i$ is bounded from above, our numerical results imply that the time $t$ needed to prepare the GHZ state in a system of size $L$, using VQCS, is $t = O(L)$. Indeed, in fig. 3, we see that the *minimum* amount of time $T = \min_{(\gamma,\beta)}\left[\sum_i^{p=L/2}(\gamma_i + \beta_i)\right]$ amongst all the solutions that we numerically found at $p = L/2$, gives an almost perfect linear trend $T \sim L$ (see Appendix A for the explicit optimal angles). In the digital setting, the circuit depth is at most $3p = 3L/2$.

We remark that various quantum circuits are known to also exactly prepare the GHZ state (for example, using a combination of Hadamard and CNOT gates, see also Appendix B). Furthermore, experimentally, GHZ states of various sizes have been prepared with high fidelity using the Mølmer-Sørensen technique [41, 42] in trapped ions. Our VQCS protocol is complementary in that it provides a uniform circuit that achieves the same result.

## 4.2 Critical state

Let us also consider the preparation of a critical state, namely the ground state of the critical 1d transverse field Ising model (TFIM) on a ring,

$$H_T := -\sum_{i=1}^{L} Z_i Z_{i+1} - \sum_{i=1}^{L} X_i. \tag{9}$$

Similarly as before, we assume the same operations $H_1 = -\sum_{i=1}^{L} X_i$, $H_2 = -\sum_{i=1}^{L} Z_i Z_{i+1}$ as before, though now we minimize the new cost function (6) using $H_T$ above. To benchmark the

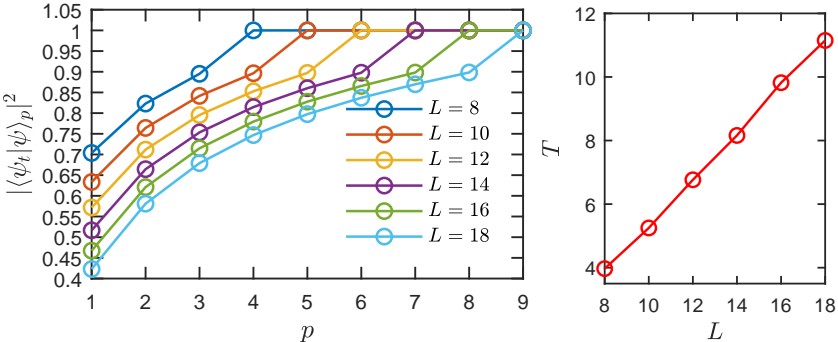

Figure 4: Preparation of critical state. (Left) Many-body overlap $|\langle\psi_t|\psi\rangle_p|^2$ of the prepared state with the target ground state of (9) found by exact diagonalization. Ones sees perfect fidelity for $p \geq L/2$. (Right) Total minimum time $T = \min_{(\gamma,\beta)}\left[\sum_i^{p=L/2}(\gamma_i + \beta_i)\right]$ required for the VQCS to produce the critical state with perfect fidelity using VQCS$_{p=L/2}$. One sees a linear trend $T \sim L$.

simulation, we compute the many-body overlap $|\langle\psi_t|\psi\rangle_p|^2$ of the prepared state $|\psi\rangle_p$ with the corresponding target state $|\psi_t\rangle$ (the ground state of (9), obtained by exact diagonalization).

Figs. 4 show the results (see Appendix C for energy optimization plots and explicit optimal angles). Surprisingly, the critical state $|\psi_t\rangle$ can also be prepared with *perfect* fidelity to machine precision using VQCS$_{p^*}$, with $p^* = L/2$. This implies once again that the time $t$ needed to prepare a critical state of this system of size $L$, exactly, goes as $t = O(L)$; we find additionally numerically that the minimum time $T$ required scales linearly with the system size as $T \sim L$.

### 4.3 Ground states of the TFIM at generic points in the phase diagram

The perfect fidelities achieved for both the GHZ and critical cases using VQCS$_{p^*}$ for $p^* = L/2$ suggest that other points $g$ in the phase diagram of the TFIM $H_{\text{TFIM}} := -\sum_{i=1}^{L} Z_i Z_{i+1} - g\sum_{i=1}^{L} X_i$, might similarly be targeted. In fact, we conjecture that, for a one-dimensional system of even $L$ spin-1/2s with periodic boundary conditions, any state produced by VQCS$_p$ for arbitrary $p$ using $H_2 = -\sum_{i=1}^{L} Z_i Z_{i+1}$ and $H_1 = -\sum_{i=1}^{L} X_i$, can also be achieved perfectly by VQCS$_{p=L/2}$. This would imply that we can indeed achieve the ground state of $H_{\text{TFIM}}$ at *any* point $g$ in the phase diagram using VQCS$_{p=L/2}$, which in particular would cover the GHZ and critical cases. In Appendix D, we provide extra details and numerical evidence to support this conjecture.

We note that the perfect fidelities achieved (to numerical precision) suggest that an analytic understanding may be possible. However, while the model and unitary gates can be mapped to free fermions [43], the minimization of the VQCS cost function maps to a nonlinear optimization problem involving an extensive number of variables, which is highly nontrivial.

### 4.4 Ground state of the Toric code

We next consider the preparation of a topologically ordered state, specifically the ground state of the $\mathbb{Z}_2$ Wen-plaquette model on a square lattice, which is unitarily equivalent to the Kitaev toric code:

$$H_T = -\sum_{i=1}^{L}\sum_{j=1}^{L} \sigma_{i,j+1}^x \sigma_{i+1,j+1}^y \sigma_{i+1,j}^x \sigma_{i,j}^y, \tag{10}$$

where we have written the Pauli matrices $(X, Y, Z)$ as $(\sigma^x, \sigma^y, \sigma^z)$ and assumed periodic boundary conditions with even $L$. Taking $H_2 = H_T$ and $H_1 = -\sum_{i=1}^{L} X_i$ in the VQCS, we find that there exists a protocol of $p = L/2$ iterations which perfectly prepares the ground state of (10).

The result can be understood from a map between the original $\sigma$ spin variables and a dual set of spin variables $\tau$s residing on the centers of plaquettes, which map the Wen plaquette model to decoupled chains of Ising models living on the diagonals. Concretely, let $\mathcal{H}_S$ be the Hilbert space subject to the $L$ constraints $\prod_{i=1}^{L} \sigma_{i,j}^x = 1$ for $j = 1, ..., L$, which is conserved under time evolution by $H_I$ and $H_X = -\sum_{i,j} \sigma_{i,j}^x$, which has $\dim \mathcal{H}_S = 2^{L^2-L}$. We now define a new set of Pauli operators $\tau$ residing on the centers of plaquettes (see also [44]); $\tau_{i,j}$ is located on the center of the plaquette with lower left corner at $(i, j)$. All operators preserving $\mathcal{H}_S$ can be rewritten in terms of $\tau$ via:

$$\begin{aligned} \tau_{i,j}^x &= \sigma_{i,j+1}^x \sigma_{i+1,j+1}^y \sigma_{i+1,j}^x \sigma_{i,j}^y, \\ \tau_{i,j}^z \tau_{i+1,j+1}^z &= \sigma_{i+1,j+1}^x, \end{aligned} \tag{11}$$

subject to the $L$ constraints $\prod_{i=1}^{L} \tau_{i,j}^x = 1$ for $j = 1, ..., L$.

As the goal is to transform the trivial product state stabilized by $H_1 = -\sum_{i=1}^{L} \sum_{j=1}^{L} \sigma_{i,j}^x$ to the topologically ordered state stabilized by $H_2$ and the two logical operators $L_1 = \prod_{i=1}^{L} \sigma_{i,i}^x$ and $L_2 = \prod_{i=1}^{L} \sigma_{i,i+1}^x$, this is equivalent in the dual language to transforming the state stabilized by

$$\left\{ -\tau_{i,j}^z \tau_{i+1,j+1}^z = -\sigma_{i+1,j+1}^x \right\}_{i=1}^{L} \tag{12}$$

and $\prod_{i=1}^{L} \tau_{i,j}^x = 1$, into the state stabilized by

$$\left\{ -\tau_{i,j}^x = -\sigma_{i,j+1}^x \sigma_{i+1,j+1}^y \sigma_{i+1,j}^x \sigma_{i,j}^y \right\}_{i=1}^{L}, \tag{13}$$

i.e. converting the GHZ state defined on each diagonal (labeled $j$) of $\tau$ spins, to the trivial product state $\bigotimes_i |+\rangle_i$. Since there exists a unitary protocol corresponding to $\text{VQCS}_{p=L/2}$ that prepares the GHZ state (shown earlier), the inverse of the protocol can be applied onto each diagonal of $\tau$ spins to achieve this result. In fact, since operators between diagonals commute, the unitaries on each diagonal can in fact be done in parallel, i.e. a global evolution, and the ground state of the toric code prepared. Moreover, the logical operator $(L_1, L_2)$ constraints are preserved at all steps (see Appendix E for a numerical illustration). The total minimum runtime $T$ needed to implement this protocol, as found earlier, scales as $T \sim L$, which we note is a lower bound derived in [38].

## 4.5 Ground state of AFM Heisenberg chain

Lastly, we consider targeting the ground states of the AFM spin-1/2 Heisenberg chains with open boundary conditions,

$$H_T = \sum_{i=1}^{L-1} \mathbf{S}_i \cdot \mathbf{S}_{i+1}. \tag{14}$$

We use in this case $H_1 = \sum_{i=1}^{L/2-1} \mathbf{S}_{2i} \cdot \mathbf{S}_{2i+1}$ and $H_2 = \sum_{i=1}^{L/2} \mathbf{S}_{2i-1} \cdot \mathbf{S}_{2i}$, whilst evaluating the expectation value of (14), i.e. the energy, as the cost function. Note that the initial state is now the product state of Bell pairs $\bigotimes_i \frac{1}{\sqrt{2}}(|\uparrow\downarrow\rangle - |\downarrow\uparrow\rangle)_{2i-1,2i}$, and that the angles can be restricted to $\gamma, \beta \in [0, 2\pi)$.

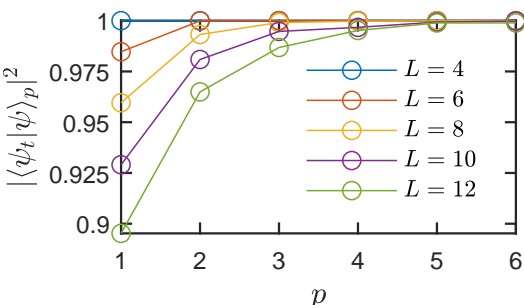

Figure 5: Fidelities in the preparation of ground states of AFM spin-1/2 Heisenberg chains of various sizes using VQCS.

Fig. 5 shows the results of the resulting fidelities, for various $L$s. We see that in this case, while the preparation of the ground states is generically not perfect at any finite $p$, the many-body fidelities are already very good for very low $p$s, at least for small system sizes, (e.g. $\sim 90\%$ at $L = 12$ and $p = 1$). This illustrates the utility and generality of VQCS in preparing non-trivial quantum states.

## 5  Conclusion

We have presented a general, efficient approach for preparing non-trivial quantum states based on VQCS, and demonstrated numerically its efficacy and efficiency in the preparation of a number of target states of interest. The main merits of this approach are its practicality for quantum simulators and its ability to improve based on feedback from the simulator. The only two requirements–time evolution by simple Hamiltonians– are realizable in synthetic quantum systems such as trapped ions and superconducting qubits. While in our examples considered we have chiefly focused on fixed point wavefunctions of certain non-trivial phases of matter described by integerable systems, we expect that the VQCS can efficiently accommodate targeting more general ground states of interacting Hamiltonians. An important question we have also addressed in Appendix F is the effect of imperfect sequences on state preparation (such as noise); we have found that the resulting infidelities in the cases studied are reasonably small for near-term simulators, and these can be further decreased using feedback.

With the ability to efficiently prepare non-trivial quantum states, various studies are possible. Their non-trivial entanglement structure could be directly measured by preparing multiple copies of the states and using recently developed protocols [45, 46]; it would be interesting to extract the central charge of the critical system or topological entanglement entropy of the toric code state. Furthermore, truncating the analytic circuit at intermediate depth allows one to prepare a state with a boundary separating toric code and a trivial paramagnet.

More generally, in addition to providing practical protocols and variational wavefunctions, the VQCS is a potential tool for addressing questions of complexity of a ground state. In the examples provided, it furnishes circuits with minimal depth scaling with size and may offer valuable guidance in determining the circuit complexity [47–49] needed to prepare various states of matter.

# Acknowledgements

We thank Daniel Gottesman, Gábor Halász, Germán Sierra, Romain Vasseur, Soonwon Choi, Shengtao Wang, Valentin Kasper, Sonika Johri and Karan Mehta for useful discussions.

**Funding information**   WWH and THH are supported by the Moore Foundation's EPiQS Initiative through Grant No. GBMF4306 and Grant No. GBMF4304 respectively.

# A   Optimal angles for preparing GHZ state at $p = L/2$

The following are the numerically found optimized set of angles $(\gamma_1, \beta_1, \cdots, \gamma_{p=L/2}, \beta_{p=L/2})$ employed by $\text{VQCS}_{p=L/2}$ which produce the GHZ state with perfect fidelity at various system sizes and with least amount of time $T = \sum_i^{p=L/2} (\gamma_i + \beta_i)$.

$L = 8, T = 4.7867$ :

$$(0.5297, 0.5243, 0.7243, 0.6151, 0.6151, 0.7243, 0.5243, 0.5297), \tag{15}$$

$L = 10, T = 6.257$:

$$(0.5814, 0.5230, 0.6360, 0.7889, 0.5993, 0.5993, 0.7889, 0.6360, 0.5230, 0.5814), \tag{16}$$

$L = 12, T = 7.651$:

$$(0.5466, 0.5452, 0.6902, 0.7212, 0.5946, 0.7276$$
$$0.7276, 0.5946, 0.7212, 0.6902, 0.5452, 0.5466), \tag{17}$$

$L = 14, T = 9.2634$:

$$(0.6513, 0.5696, 0.5841, 0.6704, 0.7633, 0.8270, 0.5660,$$
$$0.5660, 0.8270, 0.7633, 0.6704, 0.5841, 0.5696, 0.6513), \tag{18}$$

$L = 16, T = 10.6273$:

$$(0.5846, 0.5796, 0.6105, 0.7155, , 0.7966, 0.6152, 0.6373, 0.7745,$$
$$0.7745, 0.6373, 0.6152, 0.7966, 0.7155, 0.6105, 0.5796, 0.5846), \tag{19}$$

$L = 18, T = 12.096$:

$$(0.6064, 0.5232, 0.6632, 0.7780, 0.6660, 0.6302, 0.7773, 0.7133, 0.6904,$$
$$0.6904, 0.7133, 0.7773, 0.6302, 0.6660, 0.7780, 0.6632, 0.5232, 0.6064). \tag{20}$$

# B   Explicit nonuniform unitary circuit for preparing the GHZ state

We provide here an analytic example of a unitary circuit that prepares exactly the GHZ state from the product state $|+\rangle$, complementary to the VQCS scheme, which highlights the Kramers-Wannier duality. This involves a nonuniform application of various 1-site and 2-site unitary gates. We first rewrite the spin degrees of freedom in therms of Majorana fermions, via the Jordan-Wigner transformation: $\gamma_{2j-1} = Y_j \prod_{i=1}^{j-1} X_i, \gamma_{2j} = Z_j \prod_{i=1}^{j-1} X_i$ for $j$ ranging from 1 to $L$. Then $X_j = -i\gamma_{2j-1}\gamma_{2j}$ and $Z_j Z_{j+1} = i\gamma_{2j}\gamma_{2j+1}$; the product state and GHZ state thus simply correspond to the two different dimerization patterns of Majoranas. To transform from the

state with all $i\gamma_{2j-1}\gamma_{2j} = -1$ to the state with all $i\gamma_{2j}\gamma_{2j+1} = +1$, we need to sequentially exchange Majoranas pairwise ($\gamma_1 \leftrightarrow \gamma_2, \gamma_2 \leftrightarrow \gamma_3, ...$). $S = e^{\frac{i\pi}{4}i\gamma_i\gamma_j}$ is the SWAP operator which accomplishes each exchange: $S^{-1}\gamma_{i,j}S = \mp\gamma_{j,i}$. Thus, $U$ is a product of successive SWAPs, which in the spin language is

$$U = \left(\prod_{i=1}^{L-1} e^{\frac{i\pi}{4}X_{i+1}}e^{\frac{i\pi}{4}Z_iZ_{i+1}}\right)e^{\frac{i\pi}{4}X_1}. \tag{21}$$

As the last operator (when acting on $\otimes|X = 1\rangle$) contributes an overall phase and can be neglected, we have analytically found a depth $2(L-1)$ circuit relating GHZ and product states exactly; this complements the VQCS protocol discussed earlier. We note that such SWAPs were also used in [50] to transform a product state into the ground state of the Kitaev chain, .

## C  Energy optimization plot and optimal angles for preparing critical state at $p = L/2$

In Fig. 6 we present the optimal cost function given by the energy of the TFIM,

$$F_p(\boldsymbol{\gamma}, \boldsymbol{\beta}) = {}_p\langle\psi(\boldsymbol{\gamma}, \boldsymbol{\beta})|H_{\text{TFIM}}|\psi(\boldsymbol{\gamma}, \boldsymbol{\beta})\rangle_p, \tag{22}$$

used in the preparation of the critical state.

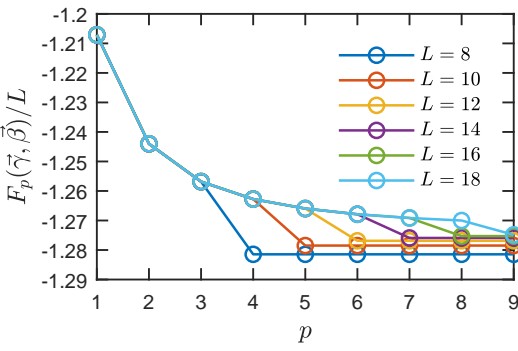

Figure 6: Preparation of critical state. Optimal cost function (22) with energy as measured by the TFIM Hamiltonian.

The following are the numerically found optimized set of angles $(\gamma_1, \beta_1, \cdots, \gamma_{p=L/2}, \beta_{p=L/2})$ employed by VQCS$_{p=L/2}$ which produce the critical state with perfect fidelity at various system sizes and with least amount of time $T = \sum_i^{p=L/2}(\gamma_i + \beta_i)$.

$L = 8, T = 3.9699$ :

$$(0.2496, 0.6845, 0.4808, 0.6559, 0.5260, 0.6048, 0.4503, 0.3180), \tag{23}$$

$L = 10, T = 5.250$:

$$(0.2473, 0.6977, 0.4888, 0.6783, 0.5559, 0.6567, 0.5558, 0.6029, 0.4598, 0.3068), \tag{24}$$

$L = 12, T = 6.7651$:

$$(0.2809, 0.6131, 0.6633, 0.4537, 0.8653, 0.4663,$$
$$0.6970, 0.6829, 0.4569, 0.7990, 0.3565, 0.4304), \tag{25}$$

$L = 14, T = 8.1604$:

$$(0.3090, 0.5710, 0.6923, 0.5648, 0.5391, 0.9684, 0.3979,$$
$$0.6852, 0.8235, 0.4474, 0.6930, 0.6465, 0.4120, 0.4104), \tag{26}$$

$L = 16, T = 9.8198$:

$$(0.3790, 0.5622, 0.5638, 0.7101, 0.9046, 0.3210, 0.6738, 0.8377,$$
$$0.8616, 0.4004, 0.5624, 0.9450, 0.5224, 0.6466, 0.4119, 0.5172), \tag{27}$$

$L = 18, T = 11.1485$:

$$(0.3830, 0.4931, 0.7099, 0.7010, 0.5330, 0.6523, 0.6887, 1.0405, 0.3083,$$
$$0.6215, 0.9607, 0.5977, 0.6209, 0.5597, 0.7850, 0.5851, 0.4132, 0.4948). \tag{28}$$

## D  A Conjecture and Numerical Support

Consider a one-dimensional system of an even number $L$ spin-1/2s with periodic boundary conditions, and consider $H_I' = -\sum_{i=1}^{L} Z_i Z_{i+1}$ and $H_X = -\sum_i X_i$. Our conjecture is that any state produced by a $\text{VQCS}_p$ protocol of arbitrary $p$ can be obtained by $\text{VQCS}_{p=L/2}$. In other words, for any $p$ and set of angles $(\boldsymbol{\gamma}, \boldsymbol{\beta}) \equiv (\gamma_1, \cdots \gamma_p, \beta_1, \cdots \beta_p)$, there exists a set of angles $(\boldsymbol{\gamma'}, \boldsymbol{\beta'}) \equiv (\gamma_1', \cdots \gamma_{L/2}', \beta_1', \cdots \beta_{L/2}')$ such that

$$e^{-i\beta_{L/2}'H_X} e^{-i\gamma_{L/2}'H_I'} \cdots e^{-i\beta_1'H_X} e^{-i\gamma_1'H_I'} |+\rangle = e^{-i\beta_p H_X} e^{-i\gamma_p H_I'} \cdots e^{-i\beta_1 H_X} e^{-i\gamma_1 H_I'} |+\rangle. \tag{29}$$

It suffices to establish this result for $p = L/2 + 1$, because one could then contract the $p = (L/2 + 1)$ VQCS unitary into a $p = L/2$ VQCS unitary, and iterate this process to achieve finally a $p = L/2$ VQCS unitary. We have tested this result for different system sizes by generating random states $|\psi^{(r)}(\boldsymbol{\gamma}, \boldsymbol{\beta})\rangle_{L/2+1}$ produced using the $\text{VQCS}_{p=L/2+1}$ protocol with random angles $(\gamma_1, \cdots \gamma_{L/2+1}, \beta_1, \cdots, \beta_{L/2+1})$, and targeting them using the protocol $\text{VQCS}_p$ for $p$ up to $L/2$. More precisely, given a random state $|\psi^{(r)}(\boldsymbol{\gamma}, \boldsymbol{\beta})\rangle_{L/2+1}$, we maximize the fidelity

$$f_p(\boldsymbol{\gamma'}, \boldsymbol{\beta'}) = \left| {}_p\langle \psi(\boldsymbol{\gamma'}, \boldsymbol{\beta'}) | \psi^{(r)}(\boldsymbol{\gamma}, \boldsymbol{\beta})\rangle_{L/2+1} \right|^2, \tag{30}$$

over $(\boldsymbol{\gamma'}, \boldsymbol{\beta'})$, where $|\psi(\boldsymbol{\gamma'}, \boldsymbol{\beta'})\rangle_p$ is the state produced by $\text{VQCS}_p$.

Figs. 7, 8 show the results. In fig. 7, we plot the typical optimal infidelity $1 - \text{Median}(f_p)$, given by the median over all realizations of random states (we have used 5000 random states and ensured convergence of the algorithm to the global minimum) against $p$, and for various $L$s. We see that a typical run of $\text{VQCS}_p$ for $p = L/2$ is able to target the input state $|\psi^{(r)}(\boldsymbol{\gamma}, \boldsymbol{\beta})\rangle_{L/2+1}$ with perfect fidelity (to machine precision), while not for $p < L/2$. The reason we do not use the mean value, is because this undesirably overly weights the contributions of numerical imprecisions in the optimization algorithm. However, to make a statement about whether $\text{VQCS}_{p=L/2}$ is able to *always* reach the target random state, we need to analyze the full *distribution* of the optimal fidelities. In fig. 8, we plot the *distribution* of the optimal fidelities for one of the system sizes considered and for various $p$s by plotting the probability distributions $P(f)$ of the optimal fidelities $f$. We find that at $p = L/2$, the distribution is singularly peaked at $f = 1$ (to machine precision), indicating that in fact, *all* realizations of random states created using $\text{VQCS}_{p=L/2+1}$ can be targeted with $\text{VQCS}_{p=L/2}$, perfectly. This is in contrast to the optimal fidelities obtained for $p < L/2$: there is some spread in the distributions,

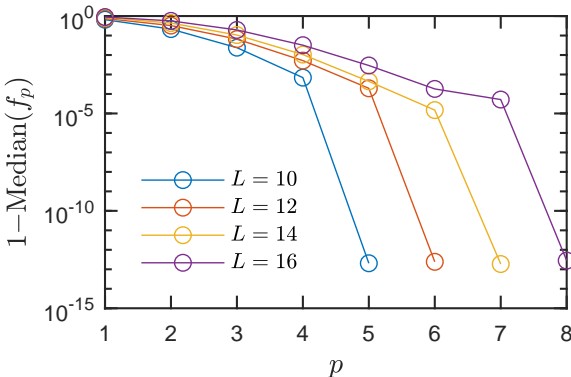

Figure 7: Typical optimal infidelity of $\text{VQCS}_p$ for $1 \leq p \leq L/2$ used to target a random state produced by $\text{VQCS}_{p=L/2+1}$ (given by the median over 5000 realizations of random states). One sees a clear dip at $p = L/2$, to a value close to machine precision (which we take to be $\sim 10^{-13}$), indicating that the $\text{VQCS}_{p=L/2}$ is able to target a random state with perfect fidelity typically.

indicating that there are instances of random states for which $\text{VQCS}_{p<L/2}$ cannot reproduce it. Thus, our numerics gives support to the conjecture that any state produced using $\text{VQCS}_{p\geq L/2+1}$ can be obtained by $\text{VQCS}_{p=L/2}$.

One important consequence of the above conjecture is that the ground state of any point in the transverse field Ising model ($H = H'_I + gH_X$ for arbitrary $g$) can be achieved with perfect fidelity by $\text{VQCS}_{p=L/2}$. This is because as the number of iterations $p$ approaches infinity, VQCS includes the trotterized adiabatic algorithm as a subset, and the latter can achieve any ground state in the phase diagram if infinite depth is permitted. Our conjecture then implies that such a protocol can be contracted to one with $p = L/2$.

As for proving the conjecture, we note that that leveraging the free fermion representation of the model, as done in [43], is a promising route. However, such a representation nonetheless involves a nonlinear (and hence nontrivial) optimization problem which we leave for future work.

# E   Numerical verification of preparation of toric code ground state

We show here numerics that verify that we can prepare using $\text{VQCS}_{p=L/2}$ the ground state of the Wen-plaquette model in the sector $(L_1, L_2) = (+1, +1)$, using the angles found previously of a $\text{VQCS}_{p=L/2}$ protocol which prepared the GHZ state. Fig. 9 shows the result for a $L \times L$ Wen-plaquette model, where $L = 4$ (so that there are only four angles $(\gamma_1, \gamma_2, \beta_1, \beta_2)$ employed by the VQCS protocol). We see that all plaquette operators and logical operators carry a unit expectation value in the prepared state, which indicates that we can indeed prepare the ground state of the Wen-plaquette model in the appropriate logical sector as mentioned in the main text.

Note that this sequence derived from VQCS is different from the analytic depth-$2(L-1)$ circuit (using SWAP operators) that also prepares the Wen-plaquette ground state exactly.

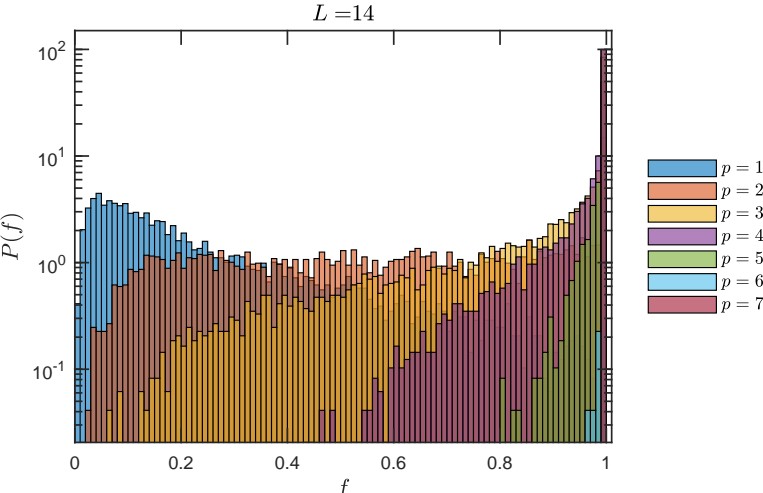

Figure 8: Probability distribution of optimal fidelities, for system size $L = 14$. For $p = L/2$, the optimal fidelities are singularly peaked at $f = 1$, indicating that *all* instances of random states produced by VQCS$_{p=L/2+1}$ can be targeted using VQCS$_{p=L/2}$ perfectly; this is in contrast to $p < L/2$ where there is some spread in the distribution, indicating that there are instances of random states for which VQCS$_{p<L/2}$ cannot target them. Probability distributions for other system sizes is qualitatively similar to one shown here.

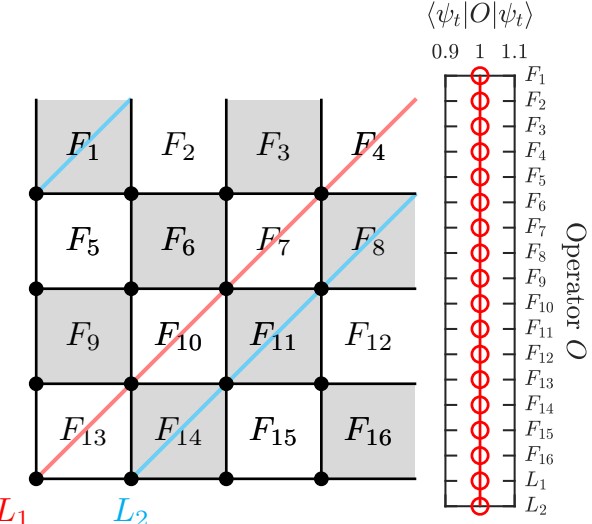

Figure 9: Numerical preparation the of Wen plaquette ground state using VQCS$_{p=L/2}$. Here $L = 4$, and we use the angles found from VQCS$_2$ that produced the GHZ state. The left plot describes the geometry of the set-up, and illustrate the plaquette operators $F_i$ which make up the Hamiltonian $H_T = -\sum_i F_i$ as well as the two logical operators $L_1$ and $L_2$ which wrap around the torus. The right plot shows the expectation value of the plaquette operators and logical operators in the state prepared by VQCS. One sees that all expectation values are $+1$ to machine precision, indicating a perfect preparation of the ground state.

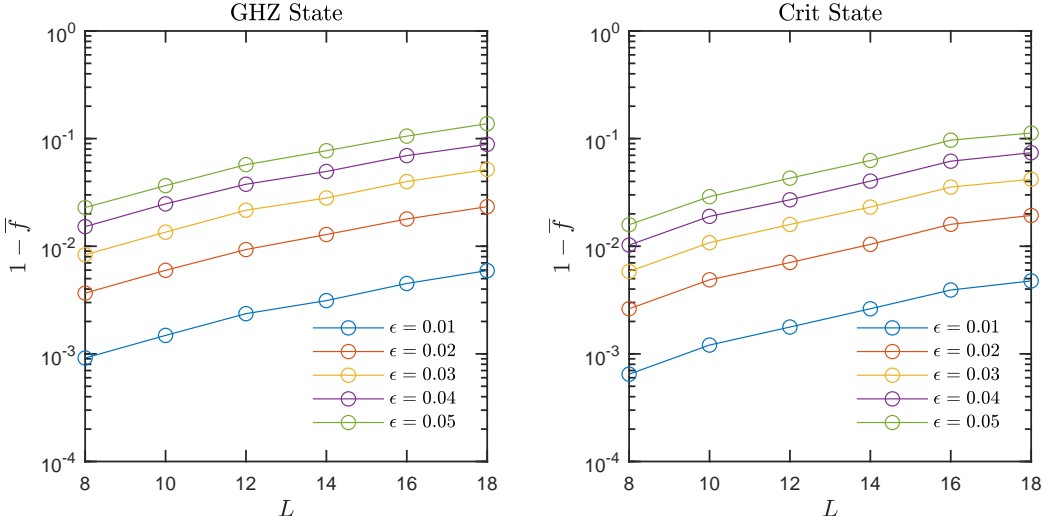

Figure 10: Effect of errors of strength $\epsilon$ on the VQCS preparation of GHZ and critical states for system size $L$. Plotted is the infidelity averaged over 1000 error realizations (denoted by the overline).

## F  Effect of errors on VQCS state preparation

To probe the sensitivity of our state preparation protocol to imperfections, we introduced random errors to the optimal angles and calculated the resulting infidelity $f = 1 - |\langle \psi_t | \psi \rangle_{L/2}|^2$ for VQCS$_{p=L/2}$, averaged over 1000 realizations of errors. Specifically, for each optimal angle $\gamma_*$, we introduce an error $\gamma = \gamma_*(1 + \epsilon R)$, where $R$ is chosen randomly from the uniform distribution $[-1, 1]$ and $\epsilon$ parameterizes the strength of error $(0.01, 0.02, 0.03, 0.04,$ or $0.05$ in our study).

Fig. 10 shows the results for both the GHZ and critical states, for various system sizes and error strengths. Although the infidelity appears to increase exponentially with $L$, we see that for experimentally accessible system sizes (on the order of ten qubits), the infidelity is small $(< 0.01$ infidelity for $\epsilon = 0.01$ in $L = 18)$.

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
