# Peer review of "Efficient variational simulation of non-trivial quantum states"

_SciPost Physics, doi:SciPost Phys. 6, 029 (2019)_

## Round 2 · Referee Report · Anonymous (Referee 1) · 2018-12-20

Strengths

  1. Novel and interesting application(s) of the quantum approximate optimization algorithm (QAOA) framework.

  2. The paper is well-written and the results and examples are presented in a clean, clear and understandable manner.

  3. The main ideas of the paper are demonstrated with examples accessible to a relatively broad audience.

Weaknesses

  1. The implementation costs resulting from the constructions of this paper - and subsequently any advantages or disadvantages of the constructions - is unclear. The scaling of the total time (sum of the algorithm parameters) is emphasized in the paper; however, a more important quantity, the scaling of the quantum circuit depth (which scales with the number of QAOA$_p$ repetitions p) is underemphasized and underexplored. While closely related, these quantities are not identical, and how these two quantities relate is not explained in the paper.

This point is elaborated on in the Requested Changes section

  1. Similarly, "various quantum circuits" are mentioned as alternatives to the applications considered in this paper, yet the implementation costs are not compared in detail nor other (dis)advantages presented.
    (An example of such a circuit is given in App. B.)

[For example, comparing the "total time" of two quantum gate-model algorithms is typically not meaningful, whereas metrics such as circuit depth, are.]

Report

The paper "Efficient preparation of non-trivial quantum states using the Quantum Approximate Optimization Algorithm" shows a novel application of QAOA to preparing quantum states, including several provided examples taken from quantum physics.

The results of this paper are of scientific significance as they relate to potential impactful uses of quantum computers. Moreover, quantum state preparation is an important paradigm in quantum computing, generally, with many applications beyond those of this paper.

The paper is generally well-written and the ideas and results are presented clearly.

Hence, modulo a few minor comments below, I am happy to recommend acceptance of this paper for publication.

Requested changes

  1. In the abstract, it is stated that the time scales as $O(L)$. It is not clear what "time" means here without some context; c.f. the related comments in this report. 1'. Is $O(L)$ what you mean here? - this implies linear or $sub-linear$ scaling - perhaps replace with $\Theta(L)$ or the phrase "scales linearly".

  2. The title of section 3 is "Quantum Approximate Adiabatic Algorithm (QAOA)" - this is confusing to the reader and should be either "Quantum Approximate Optimization Algorithm (QAOA)" or "Quantum Approximate Adiabatic Algorithm"

  3. For the benefit of the reader, the GHZ state when first defined should be given as a numbered equation.

  4. As mentioned, the paper would benefit from clarification of the relation of the stated results (such as total time) to the circuit complexity (i.e., implementation cost in terms of circuit size and depth) for the given constructions.

Consider for example the GHZ state construction, though the point here applies also to the other constructions of the paper.

The GHZ Hamiltonian (3) can be simulated in many reasonable quantum gate sets with circuit depth that does not depend on $L$, as can be the paramagnetic Hamiltonian. Hence, (i) the QAOA$_{p=L/2}$ quantum circuit would have circuit depth that scales linearly with $L$, independently of the particular algorithm parameters $\gamma,\beta$. The authors then find empirically that (ii) the total time T to achieve perfect fidelity also scales linearly with $L$. Without an explanation otherwise, facts (i) and (ii) appear to be unrelated. Fact (ii) is reported in the abstract, while fact (i) is often more significant.

Indeed, alternative algorithms for preparing this state may not have well-defined "total times" for comparison to the approach of this paper, and quantum circuit depth is typically a more fruitful metric.

  • validity: high
  • significance: good
  • originality: high
  • clarity: high
  • formatting: excellent
  • grammar: excellent

Author:  Wen Wei Ho  on 2019-01-07  [id 396]

(in reply to Report 1 on 2018-12-20)

We thank the Referee very much for the appreciation of our work, the insightful comments which will help improve our paper, and also for the recommendation for publication in SciPost.

We would like to respond to some of the Referee's main comments:

1) The Referee mentions that a more meaningful quantity than the scaling of the total time $T$ of the algorithm (which is the sum of the total angles $T = \sum_{i=1}^p (\beta_i + \gamma_i)$) is the closely related quantity, the QAOA circuit depth ($p$). We note that we have reported both metrics in the paper: they scale linearly with system size $L$. Indeed, in our paper, at least for the cases studied, the total time is related to the circuit depth as $T=O(1)$ times depth $p$ because each angle $\beta_i,\gamma_i$ are uniformly upper bounded by some constant and are hence $O(1)$ times.

However, we respectfully disagree that the QAOA-depth $p$ is the more meaningful metric. We note that the QAOA algorithm is not a 'digital' quantum circuit, that is, it is not an algorithm where gates from a fixed set (e.g. CNOT, Hadamard, etc) are chosen. Thus, treating the QAOA depth $p$ as the depth of a quantum circuit, the latter of which counts the number of times gates from this fixed set are drawn which determines the algorithm's complexity, can be misleading. This is because each layer of the QAOA iteration depends also on an analog parameter $\gamma_i$ or $\beta_i$ (which have the interpretation of runtimes, and which can be small or large). Since physical near-term quantum simulators have a limitation on how long they can be coherently run for (crudely set by the $T_2$ time), a comparison of the total time $T$ taken for the algorithm to run relative to this coherence time is a meaningful one.

As a concrete example, suppose we have two QAOA 'circuits' of similar depth $p$. In one case, say all angles $\beta_i, \gamma_i$s are very close to $0$; while in the other case say the angles are large, $\gg 0$. Then arguably the latter circuit is more 'complex', in the sense that it will accumulate more errors in a physical implementation of the algorithm. (We note that we have provided an analysis of this line of reasoning in the appendix; though this analysis ignores possible imperfections due to 'switching' between Hamiltonians in the implementation). Moreover, we note that the physical runtime is the more meaningful one for comparison with a more conventional algorithm, for example, the quantum adiabatic algorithm (QAA).

With regards to the Referee's example about the GHZ state -- we agree that facts (i) and (ii) raised are distinct, although related. Paraphrased, we find empirically that: (i) The QAOA depth to achieve perfect fidelity of the studied target states is $p = L/2$. (ii) Point (i) does not, however, determine the scaling of the total time (although we can say they are upper bounded by a constant * p, as mentioned above). We furthermore find that the total time scales as $L$. These are indeed two separate points as the Referee mentioned, though for reasons stated above, we believe (ii) to be more physically relevant.

Nevertheless, we appreciate the Referee's critique and have taken the suggestions constructively -- as per the Referee's advice, we will include a discussion similar to the one we have provided here regarding the differences in the two metrics, and to highlight the fact that points (i) and (ii) are distinct, but related, statements, in both the abstract and the main text. We believe this will help clarify the differences between the two metrics which will improve the manuscript.

2) Regarding the requested changes on the title of Section 3 (this was indeed a typo; it should read Quantum Approximation Optimization Algorithm) and the definition of the GHZ state, we will rectify that in the next iteration. We thank the Referee for pointing these out.

We thank the Referee once again for his/her comments. Sincerely, Authors

Anonymous on 2019-01-26  [id 414]

(in reply to Wen Wei Ho on 2019-01-07 [id 396])

Once again I cannot agree with the authors, and I am, quite frankly, surprised by their insistence.
Their statements appear to reflect personal opinion rather than the literature.

The authors correctly state that the total time and QAOA circuit depth are "closely related" quantities and that each angle is "uniformly upper bounded by some constant".
However, in the next paragraph, they refer to the same angles as "analog parameters..which can be small or large."
This is inconsistent with the previous statement.
Please refer to how small constants are treated in the computer science literature on algorithms.

The authors state that "the QAOA algorithm is not a 'digital' quant circuit..where gates from a fixed set".
This is not true, as clearly is clearly evidenced by the literature.
See e.g.:
[Farhi14] A quantum approximate optimization algorithm. E Farhi, J Goldstone, S Gutmann arXiv:1411.4028, 2014.
[Farhi17] Quantum algorithms for fixed qubit architectures. E Farhi, J Goldstone, S Gutmann, H Neven arXiv:1703.06199, 2017
[Had18] Quantum Algorithms for Scientific Computing and Approximate Optimization
S Hadfield arXiv:1805.03265, 2018
[Lech18] Quantum Approximate Optimization with Parallelizable Gates
W Lechner arXiv:1802.01157, 2018
[Ventu18] Compiling quantum circuits to realistic hardware architectures using temporal planners
D Venturelli, M Do, E Rieffel J Frank Quantum Science and Technology, 2018 Feb 21;3(2):025004.

Cost invariants such as circuit depth are standard in the computer science literature, for good reason, and should not be treated lightly. They provide a largely implementation-independent measure of an algorithm's cost.
If the authors wish to limit their results to "analog" platforms, that is at their loss. For digital quantum platforms, including those emerging from Google, Rigetti, etc., other factors such as connectivity are important and greatly affect the compiled low-level gate cost. Moreover, for other proposals such as fault-tolerant quantum computing schemes, the critical resource may be the number of non-clifford gates, which one expects to scale with circuit depth.
QAOA has been considered for all of these platforms, as well as for non-near-term devices;
see e.g.
[Farhi16] Quantum supremacy through the quantum approximate optimization algorithm
E Farhi, AW Harrow arXiv:1602.07674, 2016
[Jia17] Near-optimal quantum circuit for Grover's unstructured search using a transverse field
Z Jiang, EG Rieffel, Z Wang PRA, 2017

With regards to the author's paragraph "As a concrete example...", again, I cannot agree. Quantum gate model algorithms are fundamentally distinct as computational models from continuous time algorithms like the quantum adiabatic optimization; please consult the literature.
Moreover, for a gate-model algorithm such as QAOA, which is typically repeated many times in a hybrid classical loop and generates outputs probabilistically, it is not clear how noise will affect the performance of the algorithm (in contrast , e.g., to an algorithm such as Shor's where one might imagine the intermediate and final states to be much more "delicate"...).

---

## Round 3 · Referee Report · Anonymous (Referee 1) · 2019-1-27

Strengths

See previous report.

Weaknesses

See previous report.

Report

Warnings issued while processing user-supplied markup:

  • Inconsistency: Markdown and reStructuredText syntaxes are mixed. Markdown will be used.
    Add "#coerce:reST" or "#coerce:plain" as the first line of your text to force reStructuredText or no markup.
    You may also contact the helpdesk if the formatting is incorrect and you are unable to edit your text.

There seems to be much confusion regarding the remarks contained in my initial review. This reviewer is perplexed by the author's overly confident response to an initially minor comment regarding the paper.

Unfortunately the related sections of the text require further revision before I can recommend the paper for publication. Fortunately this change is relatively minor and I see no reason to delay acceptance of the paper assuming a reasonable fix is proposed. I advise the authors to review the literature. I give a few examples below.

Note that the authors have separately provided a confusing response to my initial review of the paper. I have provided an additional response, in context, attached to the same comment thread - please see the associated webforms. Some of my same points will be repeated below. I have further pasted my response at the bottom of this text.

===== Some further specific comments regarding the paper revision:

  • In the revised abstract, it is not explained what the parameter p is

-As the QAOA time parameters are each bounded by a small constant, the total time is bounded by a constant times the QAOA circuit depth p. The authors appear to have made statements both agreeing with and contradicting this statement.

  • Regarding the revisions to section 3: The authors' insistence that QAOA should be seen as an analog algorithm (only) is inexplicable. This contradicts the literature, where it is presented as a digital quantum algorithm; see e.g.: [Farhi14] A quantum approximate optimization algorithm. E Farhi, J Goldstone, S Gutmann arXiv:1411.4028, 2014. [Farhi17] Quantum algorithms for fixed qubit architectures. E Farhi, J Goldstone, S Gutmann, H Neven arXiv:1703.06199, 2017 [Had18] Quantum Algorithms for Scientific Computing and Approximate Optimization S Hadfield arXiv:1805.03265, 2018 [Lech18] Quantum Approximate Optimization with Parallelizable Gates W Lechner arXiv:1802.01157, 2018 [Ventu18] Compiling quantum circuits to realistic hardware architectures using temporal planners D Venturelli, M Do, E Rieffel J Frank Quantum Science and Technology, 2018 Feb 21;3(2):025004.

-The authors write "We note here that it is tempting to view the number of iterations p of the QAOA protocol as the ‘depth’ of a quantum circuit and relate that to its ‘complexity’." This is inconsistent with the computer science literature. Notions of algorithm complexity and gate depth are standard and important notions in both classical and quantum circuit complexity. The whole point of circuit depth is that it gives an architecture independent metric, an \textit{invariant}, which allows for algorithm evaluation.

Consider, for example, the original proposed implementation of QAOA for the MaxCut problem in [Farhi14]. The initial state can be prepared in depth 1 using Hadamard gates. The mixing operator can be applied in depth 1 using X rotation gates. The phase operator can be applied in depth proportional to the maximum graph degree using ZZ-rotations. So in this case the gate depth follows directly from the QAOA circuit depth.

Nevertheless, at the level of compilation to primitive quantum gates, different architectures may require additional auxiliary interactions (e.g., swap gates to interact distant qubits, or non-clifford gates in fault tolerant settings), so the cost implementing of the QAOA operators with bounded angles is often bounded by a constant.

In different quantum algorithm such as e.g. Hamiltonian simulation the time parameter is not bounded by a constant and becomes an input complexity parameter, but this is not the case here.

===== Response to the authors' comments on the initial review of their paper: =====

Once again I cannot agree with the authors, and I am, quite frankly, surprised by their insistence. Their statements appear to reflect personal opinion rather than the literature.

The authors correctly state that the total time and QAOA circuit depth are "closely related" quantities and that each angle is "uniformly upper bounded by some constant". However, in the next paragraph, they refer to the same angles as "analog parameters..which can be small or large." This is inconsistent with the previous statement. Please refer to how small constants are treated in the computer science literature on algorithms.

The authors state that "the QAOA algorithm is not a 'digital' quant circuit..where gates from a fixed set". This is not true, as clearly is clearly evidenced by the literature. See e.g.: [Farhi14] A quantum approximate optimization algorithm. E Farhi, J Goldstone, S Gutmann arXiv:1411.4028, 2014. [Farhi17] Quantum algorithms for fixed qubit architectures. E Farhi, J Goldstone, S Gutmann, H Neven arXiv:1703.06199, 2017 [Had18] Quantum Algorithms for Scientific Computing and Approximate Optimization S Hadfield arXiv:1805.03265, 2018 [Lech18] Quantum Approximate Optimization with Parallelizable Gates W Lechner arXiv:1802.01157, 2018 [Ventu18] Compiling quantum circuits to realistic hardware architectures using temporal planners D Venturelli, M Do, E Rieffel J Frank Quantum Science and Technology, 2018 Feb 21;3(2):025004.

Cost invariants such as circuit depth are standard in the computer science literature, for good reason, and should not be treated lightly. They provide a largely implementation-independent measure of an algorithm's cost. If the authors wish to limit their results to "analog" platforms, that is at their loss. For digital quantum platforms, including those emerging from Google, Rigetti, etc., other factors such as connectivity are important and greatly affect the compiled low-level gate cost. Moreover, for other proposals such as fault-tolerant quantum computing schemes, the critical resource may be the number of non-clifford gates, which one expects to scale with circuit depth. QAOA has been considered for all of these platforms, as well as for non-near-term devices; see e.g. [Farhi16] Quantum supremacy through the quantum approximate optimization algorithm E Farhi, AW Harrow arXiv:1602.07674, 2016 [Jia17] Near-optimal quantum circuit for Grover's unstructured search using a transverse field Z Jiang, EG Rieffel, Z Wang PRA, 2017

With regards to the author's paragraph "As a concrete example...", again, I cannot agree. Quantum gate model algorithms are fundamentally distinct as computational models from continuous time algorithms like the quantum adiabatic optimization; please consult the literature. Moreover, for a gate-model algorithm such as QAOA, which is typically repeated many times in a hybrid classical loop and generates outputs probabilistically, it is not clear how noise will affect the performance of the algorithm (in contrast , e.g., to an algorithm such as Shor's where one might imagine the intermediate and final states to be much more "delicate"...).

Requested changes

See the above report.

---

## Round 3 · Author Response

Dear Editors,

With this reply letter, we would like to submit a revised version of our manuscript "Efficient preparation of non-trivial quantum states using the Quantum Approximate Optimization Algorithm" for your consideration for publication in SciPost. We thank you for arranging the review of our manuscript.

We thank the Referee once again for the careful consideration, and also for the insightful comments which helped improve our work. We are happy to learn of the Referee's appreciation of the impact of our work ("novel and interesting applications", "accessible to a relatively broad audience"), and also of the recommendation of acceptance of the paper to SciPost.

In the revised version of the manuscript, we have effected all of the changes requested by the Referee and as detailed in our Reply previously. In particular, we have added a discussion in Sec 3 regarding the relation between the QAOA iteration depth p and the total runtime t of the algorithm, and explained why the latter is the more meaningful of the two metrics in terms of implementation costs. In doing so, we believe we have adequately addressed the Referee's main comment that the 'circuit depth' of the QAOA is the more fruitful metric; we humbly disagree with this point. In a nutshell (and as elaborated upon in our previous reply to the Refereee), as the QAOA is envisioned to be run on noisy near-term analog quantum simulators, the total runtime to implement the algorithm (relative to the noise) is an important metric to gauge its success. We emphasize that the QAOA is not a digital quantum circuit comprised of gates from a fixed gate set, and thus, it can be misleading to view the QAOA iteration number p as equivalent to the depth of some quantum circuit. We thank the Referee for bringing up this potentially confusing point though, and we believe the changes we have made will have clarified this issue sufficiently.

We hope that with these changes, our improved manuscript will now meet the high standards for publication in SciPost. Thank you for your consideration, and please let us know if any additional information or material could be useful.

Sincerely,
Authors

---

## Round 3 · List of Changes

1. With respect to the Refere's comment that the statement regarding scaling of the time with system size is not clear in the abstract -- we have accordingly modified the abstract to explicitly report both that (i) the number of QAOA iterations to achieve perfect fidelity is p = L/2 and that (ii) the minimum physical runtimes T needed, scales linearly as T ~ L.

  2. With regards to the title of Section 3 -- this is indeed a typo and we have corrected it to "Quantum Approximate Optimization Algorithm".

  3. As per the Refere's suggestion, we have given the GHZ state as a numbered equation [Eq. (1) in the revised version] when first defining it.

  4. With regards to clarification on the two metrics of the implementation costs -- the QAOA iterations p and the total runtime t -- as argued both in our initial Reply to the Referee and in the Reply letter accompanying the resubmission, we believe the total physical runtime t is actually the more meaningful metric (relative to the noise of a near-term quantum simulator), in contrary to the Referee's comments. We also believe that it is misleading to view the iteration depth p as the depth of a digital quantum circuit, as the QAOA is not the latter -- the QAOA explicitly depends on analog parameters $\beta_i, \gamma_i$ which are the runtimes to implement each layer $H_I, H_X$.

We have sought to clarify these points better through a number of changes in the revised manuscript: i) We have removed references to 'p' as the 'depth' of the QAOA protocol, to prevent misunderstanding of this parameter as the depth of a quantum circuit (it is not). Instead, we call it the number of iterations of the QAOA protocol.

ii) As per the Referee's request, we have added a discussion in Sec 3 regarding the relation of the two metrics p and total runtime t, and explained why t is the more meaningful quantity in light of the QAOA being envisioned to be run on noisy, intermediate, analog quantum simulators. We have also emphasized how the QAOA is not a digital quantum circuit, and provided a reference [Quantum 2, 79 (2018)] elucidating the difference.

iii) In the examples of target states studied, we have made explicit the separate points which we found numerically--as noted by the Referee, that (a) the target states studied can be prepared perfectly at p = L/2 (which implies that the total physical runtimes t = O(L) as the angles are bounded from above); and (b) that the Minimum total runtimes T are found to scale linearly with L.

iv) Related to point iii) above, we have been careful to refer to t as the runtime of a given QAOA protocol (of some fixed p), and T as the {\it minimum} runtimes of over all such t's. Equations defining t and T are given in the main text.

---

## Round 4 · Author Response

We deeply apologize for the confusion regarding the relation of our considered state preparation protocol with 'circuit depth', its digital/analog nature etc., and believe it is a misunderstanding caused on our part by certain wording and phrasing.

The state preparation protocol we consider is a hybrid variational quantum-classical approach that utilizes the resources of a quantum simulator and a classical computer in an iterative fashion, in order to produce a nontrivial quantum state. It belongs to a large class of protocols which have at its core a variational working principle, see Sec D of https://arxiv.org/abs/1509.04279 for example for a very general description. Naturally, it bears large similarity to the QAOA used to approximately solve classical optimization problems, which is indeed introduced as a gate-model algorithm as the Referee emphasized. However in our case we want to view the protocol we studied more in the context of a variational approach that can be implemented in either a gate-model setting or an analog simulator setting. In the latter case of analog quantum simulation (for example in a present day non-universal trapped ion simulator or neutral Rydberg atom simulator) for which this variational quantum-classical protocol can be run, the notion of gates is less explicit, which was why we had focused less on the 'circuit-depth' and more on the run-time in the previous versions of our paper.

Needless to say, we were inspired much by the works regarding QAOA, and had thus labeled the protocol we considered 'QAOA_p'. However we believe this naming unfortunately amounted in a large part to the confusion that arose, because as mentioned above, our aim was not to study a gate-model algorithm but rather the state preparation protocol as a general, variational approach. This led us to make the erroneous statement that the 'QAOA' is not a digital quantum algorithm (the QAOA as introduced by Farhi is, which the Referee rightly points out it is; we apologize for this). Furthermore, in our replies, we also mistakenly took the parameter p as the 'circuit-depth' of a QAOA circuit, which might have led to even more confusion when the Referee said we made contradictory statements with regards to the total time and the QAOA circuit depth (the objects we are referring to are presumably different). Lastly, another aspect which might have led to misunderstanding could have been in the specific examples considered of the GHZ state and the critical state of the TFIM. In these cases, the Ising interaction used in the protocol can be decomposed into elementary 2-site gates Z_i Z_{i+1}, in exact fashion as the QAOA as analyzed by [Farhi14, arXiv:1411.4028], which can be implemented in a digital quantum simulator. Here an immediate quantum circuit interpretation is indeed possible. However, more generally, for generic target Hamiltonians whose local terms are not mutually commuting, decomposition of our protocol into local gates is much more complicated, and it is not the aim of the paper to study the cost of this.

In response to these issues, we have accordingly modified our paper. To eliminate the confusion that arose, two of our most notable changes include, 1) not referring to the protocol as 'QAOA' but rather a hybrid variational quantum-classical simulation which we call 'VQCS'. This change is reflected in the title as well as the main text. 2) clarifying the setting in which this algorithm is envisioned to be run on -- either a digital quantum simulator or an analog quantum simulator, and made explicit the quantum resources available that can be implemented. In the case of a digital quantum simulator where evolution is performed by gates resulting in a quantum circuit for the GHZ and critical Ising cases, we have accordingly quoted the circuit depth in a manner similar to the analysis of [Farhi14]. We have also removed the discussion about the QAOA not being a digital quantum algorithm. In making these minor but important structural changes (the concrete analytical and numerical results of our paper remained unchanged), we believe that our main message is now clearer and we hope that the confusion with regards to the relation to QAOA or circuit depth is now resolved.

We emphasize that we agree with the Referee on the points that i) the QAOA is introduced in the literature as a digital quantum algorithm; ii) circuit depths as cost invariants are standard in computer science, and are important in the evaluation of an algorithm cost. It is just that the main point of our reply and our revision of the manuscript is that we would like to view the approach we consider as a general variational approach and not strictly a gate-model algorithm, and which is implementable in either an analog or digital setting. We thank the Referee very much for bringing this up; we hope we have clarified the matter and hope the paper is now suitable for publication.

---

## Round 4 · List of Changes

1. As mentioned in the reply, to reflect our study of the state preparation protocol as a variational approach that can be implemented in either analog or digital quantum simulator settings (and which is not necessarily a gate-model algorithm) we have called it 'Variational Quantum-Classical Simulation' (VQCS) instead of 'QAOA'. We hope this will reduce the confusion that arose with regard to our previous revisions and reply on the properties of QAOA as introduced by Farhi et al (which is indeed a gate-model algorithm). This change is reflected in the title as well as the main text.

  2. To clarify and make explicit the context in which the protocol is envisioned for, we have added a discussion on the platforms that it can be run in (both digital and analog simulator settings), as well as the quantum resources available in both settings, in Sec 3.

  3. In the case of the transverse field Ising model when the interactions can be implemented easily in a digital quantum simulator in terms of elementary 2-qubit gates Z_i Z_{i+1} and 1-qubit rotations X_i, and where a quantum circuit interpretation is indeed possible, we have quoted the circuit depth accordingly in a manner similar to that of [Farhi14, arXiv:1411.4028].

  4. We have completely the discussion on QAOA being an analog algorithm and on the relations between its circuit depth and complexity; as mentioned above, we view our VQCS approach as a general variational approach for which a gate-model interpretation might not always be possible.

---

## Editorial Decision

published